# GRAPH LAYOUTS AND GRAPH CONTRASTIVE LEARNING VIA NEIGHBOUR EMBEDDINGS

## ABSTRACT

In node-level graph representation learning, there are two distinct paradigms. One is known as *graph layouts*, where nodes are embedded into 2D space for visualization purposes. Another is *graph contrastive learning*, where nodes are parametrically embedded into a high-dimensional vector space based on node features. In this work, we show that these two paradigms are intimately related, and that both can be successfully approached via neighbour embedding methods. First, we introduce *graph t-SNE* for two-dimensional graph drawing, and show that the resulting layouts outperform all existing algorithms in terms of local structure preservation, as measured by $k$NN classification accuracy. Second, we introduce *graph contrastive neighbor embedding (graph CNE)*, which uses a fully-connected neural network (MLP) to transform graph node features into an embedding space by optimizing the contrastive InfoNCE objective. We show that graph CNE, while being conceptually simpler than most existing graph contrastive learning methods, produces competitive node representations and outperforms state-of-the-art MLP-based methods in terms of linear classification accuracy.

## 1 INTRODUCTION

Many real-world datasets, ranging from molecule structure to citation networks come in form of graphs. As graphs are abstract objects consisting of a set of nodes $\mathcal{V}$ and a set of edges $\mathcal{E}$, graph representation learning, i.e. embedding graph nodes into a vector space $\mathbb{R}^d$, is a popular approach in machine learning. Traditionally, a distinction is made between *graph layout* (or graph drawing) methods, which embed nodes into $\mathbb{R}^2$ for visualization purposes, and *graph contrastive learning* methods, which use higher-dimensional embeddings more suitable for downstream analysis, such as classification or clustering.

For a graph $G = (\mathcal{V}, \mathcal{E})$, graph layout methods usually only take into account its structure and obtain the layout by pulling together connected nodes. In contrast, graph contrastive learning (GCL)

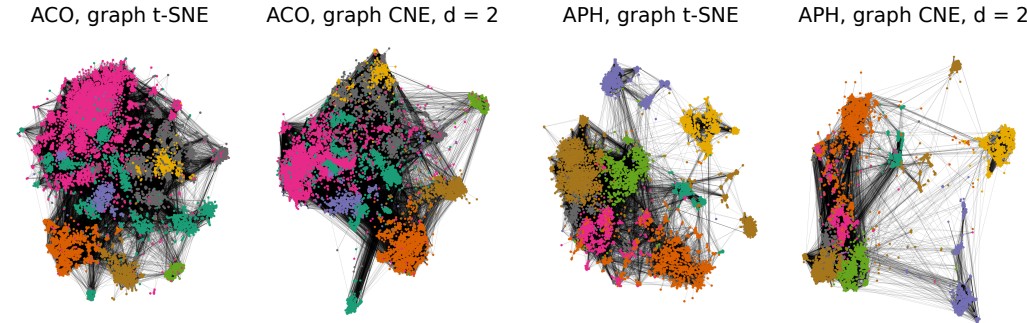

ACO, graph t-SNE     ACO, graph CNE, d = 2     APH, graph t-SNE     APH, graph CNE, d = 2

Figure 1: 2D embeddings of the Amazon Computer (ACO) and Photo (APH) datasets, obtained using our graph $t$-SNE and graph CNE. Graph $t$-SNE is a graph layout method. Graph CNE is a graph contrastive learning method, mapping node features to 2D (or other dimensionality, e.g. 128D, see Section 6) using a neural network. Embeddings were aligned using Procrustes rotation.

methods typically use node features $\mathbf{X}$ of size $n \times D$ where $n = |\mathcal{V}|$ and employ a neural network, usually a graph convolutional network (GCN) (Kipf & Welling, 2017), for the $\mathbf{R}^D \to \mathbf{R}^d$ mapping. GCL methods also pull connected nodes together, sometimes explicitly through their loss function, but also implicitly through the GCN architecture (Trivedi et al., 2022; Wang et al., 2023; Guo et al., 2023).

Recent work (Kruiger et al., 2017; Zhu et al., 2020a; Zhong et al., 2023; Böhm et al., 2022) pointed out deep connections between graph layout and neighbor embedding algorithms such as $t$-SNE (Van der Maaten & Hinton, 2008) or UMAP (McInnes et al., 2018), which are based on neighborhood preservation. In parallel, another line of work explored connections between neighbor embeddings and contrastive learning (Damrich et al., 2022; Böhm et al., 2023; Hu et al., 2023). This raises the question to what extent neighbor embedding and contrastive neighbor embedding algorithms (see Section 3) can be useful for graph representation learning.

In this work, we answer this question. We introduce a novel graph layout algorithm, graph $t$-SNE (Figure 1), and show that it strongly outperforms existing methods. We also introduce a novel, augmentation-free, GCL algorithm, graph CNE (Figure 1), based on the framework for contrastive neighbor embeddings, and show that it reaches competitive GCL performance without using GCNs. Conceptually, we present a single coherent framework for node-level graph representation learning, tying together graph layouts, graph contrastive learning, and neighbor embeddings.

## 2 RELATED WORK

**Graph layouts** Graph layout algorithms have traditionally been based on spring models, where every connected pair of nodes feels an attractive force $F_a$ and all pairs of nodes feel a repulsive force $F_r$ (*force-directed graph layouts*). Many algorithms can be written as $F_a = d_{ij}^a$ and $F_r = d_{ij}^r$ (Noack, 2007), where $d_{ij}$ is the embedding distance between nodes. For example, Fruchterman–Reingold algorithm, also known as FDP, uses $a = 2, r = -1$ (Fruchterman & Reingold, 1991); ForceAtlas2 uses $a = 1, r = -1$ (Jacomy et al., 2014); LinLog uses $a = 0, r = -1$ (Noack, 2007). Efficient implementations can be based on Barnes–Hut approximation of the repulsive forces, as in SFDP (Hu, 2005). Relationship to neighbour embeddings was discussed by Böhm et al. (2022).

**Graph layouts inspired by $t$-SNE** Several recent graph layout algorithms have been inspired by neighbor embeddings. tsNET (Kruiger et al., 2017) applied modified version of $t$-SNE to the pairwise shortest path distances between all nodes. DRGraph (Zhu et al., 2020a) made tsNET faster by using negative sampling (Mikolov et al., 2013). $t$-FDP (Zhong et al., 2023) suggested custom $F_a$ and $F_r$ forces inspired by $t$-SNE and adopted interpolation-based approximation of Linderman et al. (2019). Below we will show that our graph $t$-SNE outperforms both DRGraph and $t$-FDP. Finally, Leow et al. (2019) also suggested an algorithm called 'graph $t$-SNE', that used a graph convolutional network (Kipf & Welling, 2017) to build a parametric mapping optimizing a combination of $t$-SNE losses on node features and on shortest graph distances; it has almost no relation to our graph $t$-SNE.

**Node-level graph contrastive learning** The basic principle behind contrastive learning is to learn data representation by contrasting pairs of observations that are similar to each other (positive pairs) with those that are dissimilar to each other (negative pairs). In computer vision, positive pairs are generated via data augmentation, e.g. in SimCLR (Chen et al., 2020). Graph contrastive learning (GCL) requires node features (as input to the network) and can be graph-level or node-level, depending on whether representations are obtained for a set of graphs or for the set of nodes of a single graph. Graph-level GCL is based on graph augmentations, such as node dropping or edge perturbation, e.g. in GraphCL (You et al., 2020). Prominent examples of node-level GCL algorithms that are also based on graph augmentations include GRACE (Zhu et al., 2020b), GCA (Zhu et al., 2021), MVGRL (Hassani & Khasahmadi, 2020), DGI (Velickovic et al., 2019), BGRL (Thakoor et al., 2021), CCA-SSG (Zhang et al., 2021), etc. All of them use graph convolutional networks (GCN) to create graph embeddings.

**Augmentation-free node-level GCL** A general problem with domain-agnostic graph augmentations is that they can have unpredictable effects on graph semantics (Trivedi et al., 2022), as even minor augmentations can potentially result in a semantically different graph. This motivated development of augmentation-free GCL methods. Here positive pairs are pairs of nodes that are located

close to each other in terms of graph distance. AFGRL (Lee et al., 2022) and AF-GCL (Li et al., 2023) treat nodes with small shortest path distance as candidate positives, and use $k$ nearest neighbors in GCN-based node representations to select actual positives. Local-GCL (Zhang et al., 2022) uses all first-order graph neighbors as positives, and employs random Fourier features to approximate $\mathcal{O}(n^2)$ repulsive forces. All of these methods are also based on the GCN architecture.

## 3 BACKGROUND

### 3.1 NEIGHBOR EMBEDDINGS

Neighbor embeddings are a family of methods aiming to embed $n$ observations from some high-dimensional metric space $\mathcal{X}$ into a lower-dimensional (usually two-dimensional) vector space $\mathbb{R}^d$, such that neighborhood relationships between observations are preserved in the embedding space. Typically, $\mathcal{X}$ is another real-valued space $\mathbb{R}^p$, with $d \ll p$. We denote the original vectors as $\mathbf{x}_i \in \mathbb{R}^p$ and the embedding vectors as $\mathbf{y}_i \in \mathbb{R}^d$.

One of the most popular neighbor embedding methods, $t$-distributed stochastic neighbor embedding ($t$-SNE; Van der Maaten & Hinton, 2008) is an extension of the stochastic neighbor embedding (SNE) originally suggested by Hinton & Roweis (2002). $t$-SNE minimizes the Kullback-Leibler divergence between the high-dimensional and low-dimensional *affinities* $p_{ij}$ and $q_{ij}$:

$$\mathcal{L} = \mathrm{KL}(\mathbf{P} \, \| \, \mathbf{Q}) = \sum_{ij} p_{ij} \log \frac{p_{ij}}{q_{ij}}. \tag{1}$$

Both affinity matrices are defined to be symmetric, positive, and to sum to 1. In the original algorithm, $\mathbf{P}$ was computed using adaptive Gaussian kernels, but almost the same results can be obtained simply by normalizing and symmetrizing the $k$NN graph adjacency matrix $\mathbf{A}$ (Böhm et al., 2022):

$$\mathbf{P} = \frac{\mathbf{A}/k + \mathbf{A}^\top/k}{2n}. \tag{2}$$

Here $\mathbf{A}$ has element $a_{ij} = 1$ if $\mathbf{x}_j$ is within $k$ nearest neighbors of $\mathbf{x}_i$. Reasonable values of $k$ typically lie between 10 and 100. Low-dimensional affinities $\mathbf{Q}$ are defined in $t$-SNE using a $t$-distribution kernel with one degree of freedom, also known as the Cauchy kernel:

$$q_{ij} = \frac{(1 + \|\mathbf{y}_i - \mathbf{y}_j\|^2)^{-1}}{\sum_{k \neq l}(1 + \|\mathbf{y}_l - \mathbf{y}_k\|^2)^{-1}}. \tag{3}$$

The original SNE algorithm used Gaussian kernel instead of Cauchy, which led to worse results when embedding high-dimensional data (Kobak et al., 2019).

Even though it is usually not presented like that, $t$-SNE can be thought of as a graph layout algorithm for $k$NN graphs, in particular after the reformulation in Equation 2. During optimization, neighboring nodes (sharing an edge) feel attraction, whereas all nodes feel repulsion, arising through the normalization in Equation 3. In practice, $t$-SNE optimization can be accelerated by an approximation of the repulsive force field based on the Barnes–Hut algorithm (Van Der Maaten, 2014; Yang et al., 2013) or on interpolation (Linderman et al., 2019).

### 3.2 CONTRASTIVE NEIGHBOR EMBEDDINGS

The contrastive neighbor embedding (CNE) algorithm (Damrich et al., 2022) is a flexible framework that also operates on the $k$NN graph of the data, and optimizes the embedding in order to place connected nodes closer together than unconnected pairs of nodes. Damrich et al. (2022) considered three different loss functions: NCE (noise-contrastive estimation) (Gutmann & Hyvärinen, 2010), InfoNCE (Jozefowicz et al., 2016; Oord et al., 2018), and negative sampling (Mikolov et al., 2013). These loss functions are called *contrastive* because they are based on contrasting edges and non-edges in the same mini-batch, and do not require a global normalization like in Equation 3. Using NCE and InfoNCE in CNE approximates $t$-SNE.

Damrich et al. (2022) also considered *parametric* embeddings, where a neural network (usually a fully-connected network) is trained to produce embedding vectors $\mathbf{y}_i = f(\mathbf{x}_i)$ using one of the

Table 1: Benchmark datasets. Columns: number of nodes in the largest connected component, number of undirected edges, edges/nodes ratio, number of node classes, feature dimensionality.

| Dataset | Abbr. | Nodes | Edges | E/N | Classes | Dim. |
|---|---|---|---|---|---|---|
| CiteseerGraphDataset | CSR | 2 120 | 3 679 | 1.7 | 6 | 3 703 |
| CoraGraphDataset | COR | 2 485 | 5 069 | 2.0 | 7 | 1 433 |
| AmazonCoBuyPhotoDataset | APH | 7 487 | 119 043 | 15.9 | 8 | 745 |
| AmazonCoBuyComputerDataset | ACO | 13 381 | 245 778 | 18.4 | 10 | 767 |
| PubmedGraphDataset | PUB | 19 717 | 44 324 | 2.2 | 3 | 500 |
| ogbn-arxiv | ARX | 169 343 | 1 157 799 | 6.8 | 40 | 128 |

loss function listed above. This allows to embed new observations that have not been part of the training process. In contrast, *non-parametric* embeddings optimize $\mathbf{y}_i$ vectors directly, without any $f(\cdot)$ function. Together, this yields six combinations, called parametric/non-parametric NC-$t$-SNE, InfoNC-$t$-SNE, and Neg-$t$-SNE. Damrich et al. (2022) showed that Neg-$t$-SNE is equivalent to UMAP (McInnes et al., 2018), while NC-$t$-SNE was first suggested by Artemenkov & Panov (2020) as NCVis.

In this work we will only use the InfoNCE loss function, defined for one graph edge $ij$ (*positive pair*) as

$$\ell(i, j) = -\log \frac{q_{ij}}{q_{ij} + \sum_{k=1}^{m} q_{ik}}, \tag{4}$$

where the sum in the denominator is over $m$ *negative pairs* $ik$ where $k$ can be drawn from all nodes in the same mini-batch apart from $i$ and $j$. One mini-batch consists of $b$ graph edges, and hence contains $2b$ nodes. Therefore, for a given batch size $b$, the maximal value of $m$ is $2b - 2$. The larger the $m$, the closer InfoNC-$t$-SNE is to $t$-SNE (Damrich et al., 2022). The $q_{ij}$ affinities do not need to be normalized and are defined simply as

$$q_{ij} = (1 + \|\mathbf{y}_i - \mathbf{y}_j\|^2)^{-1}. \tag{5}$$

It is easy to see that InfoNCE loss will aim to make $q_{ij}$ large if $ij$ is a positive pair and small if it is a negative one.

When using high-dimensional embedding space, e.g. $d = 128$ instead of $d = 2$, it makes sense to define $q_{ij}$ using the Gaussian kernel transformation of the cosine distance (Damrich et al., 2022; Böhm et al., 2023):

$$q_{ij} = \exp\big(\mathbf{y}_i^\top \mathbf{y}_j / (\|\mathbf{y}_i\| \cdot \|\mathbf{y}_j\|)/\tau\big) = \text{const} \cdot \exp\Big(-\Big\|\frac{\mathbf{y}_i}{\|\mathbf{y}_i\|} - \frac{\mathbf{y}_j}{\|\mathbf{y}_j\|}\Big\|^2 \Big/ (2\tau)\Big), \tag{6}$$

where $\tau$ is called the *temperature* (by default, $\tau = 0.5$). Together with Equation 5, this gives the same loss function as used in SimCLR (Chen et al., 2020), a popular contrastive learning algorithm in computer vision. The only difference is that instead of $k$NN edges, SimCLR uses pairs of augmented images as positive pairs.

## 4 EXPERIMENTAL SETUP

**Datasets** We used six publicly available graph datasets (Table 1). All datasets were retrieved from the Deep Graph Library (Wang et al., 2019), except `ogbn-arxiv`, which was retrieved from the Open Graph Benchmark (Hu et al., 2020). Each dataset was treated as an unweighted undirected graph, where each node has a class label and a feature vector (typically a word embedding vector of some descriptive text about the node, such as a product review). We restricted ourselves to graphs with labeled nodes in order to use classification accuracy as the performance metric. We also restricted ourselves to graphs with feature vectors in order to use both non-parametric and parametric embeddings. In all datasets we used only the largest connected component, and excluded all self-loops if present, using `NetworkX` (Hagberg et al., 2008) functions `connected_components` and `selfloop_edges`.

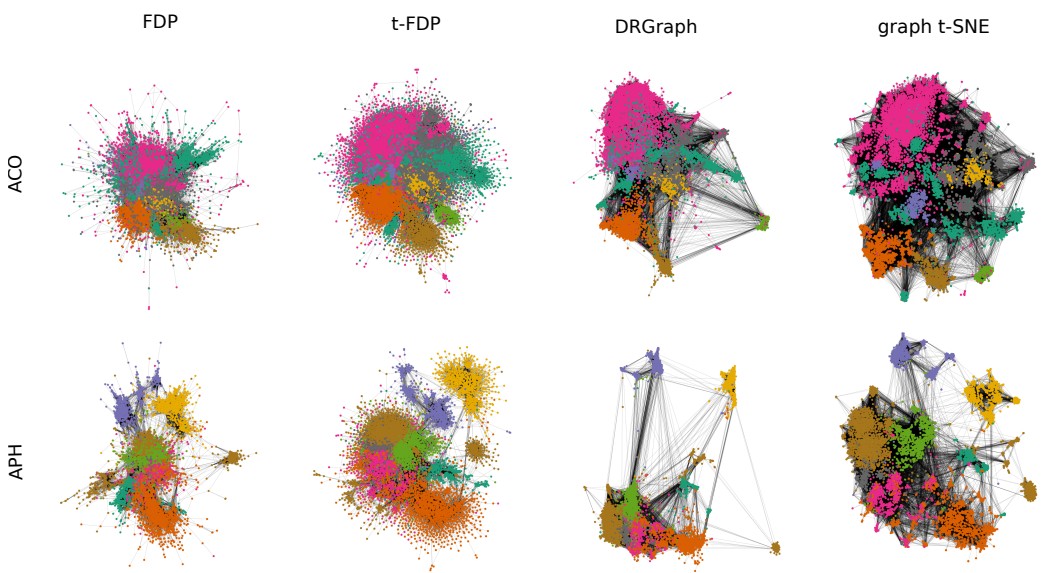

Figure 2: Embeddings of the ACO and APH datasets obtained using FDP (Fruchterman & Reingold, 1991), DRGraph (Zhu et al., 2020a), and $t$-FDP (Zhong et al., 2023), and our graph $t$-SNE. Embeddings in each row were aligned using Procrustes rotation. See Figure A.3 for all six datasets.

**Performance metrics** We evaluated the performance of our methods using three performance metrics: $k$-nearest-neighbors ($k$NN) recall, $k$NN classification accuracy, and, for high-dimensional embeddings, linear classification accuracy.

The $k$NN recall quantifies how well local node neighborhoods are preserved in the embedding. We defined it as the average fraction of each node's graph neighbors that are among the node's nearest neighbors in the embedding:

$$\text{Recall} = \frac{1}{|\mathcal{V}|} \sum_{i=1}^{|\mathcal{V}|} \frac{\left| N_G[i] \cap N_{E,k_i}[i] \right|}{k_i},$$

where $|\mathcal{V}|$ is the number of nodes in the graph, $N_G[i]$ is the set of node $i$'s graph neighbors, $N_{E,k}[i]$ denotes the set of node $i$'s $k$ Euclidean nearest neighbors in the embedding space, and $k_i = |N_G[i]|$ is the number of node $i$'s graph neighbors. This metric is similar to what is commonly used in the literature to benchmark graph layout algorithms (Kruiger et al., 2017; Zhu et al., 2020a; Zhong et al., 2023), and so is our main metric for measuring graph layout quality.

The $k$NN classification accuracy quantifies local class separation in the embedding. To calculate $k$NN accuracy, we split all nodes into a training (2/3 of all nodes) and a test set (1/3 of all nodes), and used the `sklearn.neighbors.KNeighborsClassifier` with $k = 10$ (Pedregosa et al., 2011). Of note, we used the train/test split only for training the classifier but not for computing the graph embedding itself. We used `sklearn.preprocessing.StandardScaler` to standardize all features based on the training set.

For graph CNE with $d = 128$, trained using cosine distance, we experimented with using cosine-distance-based $k$NN recall and accuracy, but found that it gave very close results to the Euclidean-distance-based $k$NN evaluations (all differences below 1 percentage point).

For linear accuracy we used the `sklearn.linear_model.LogisticRegression` class with no regularization (`penalty=None`) and otherwise default parameters, and the same train/test split. Features were standardized using `StandardScaler`.

**Computing environment** All computations were performed on a remote computing server with an Intel Xeon Gold CPU with 16 double-threaded 2.9 Ghz cores, 384 GB of RAM, and an NVIDIA RTX A6000 GPU. GPU training was used for CNE models but not for $t$-SNE. Computation times

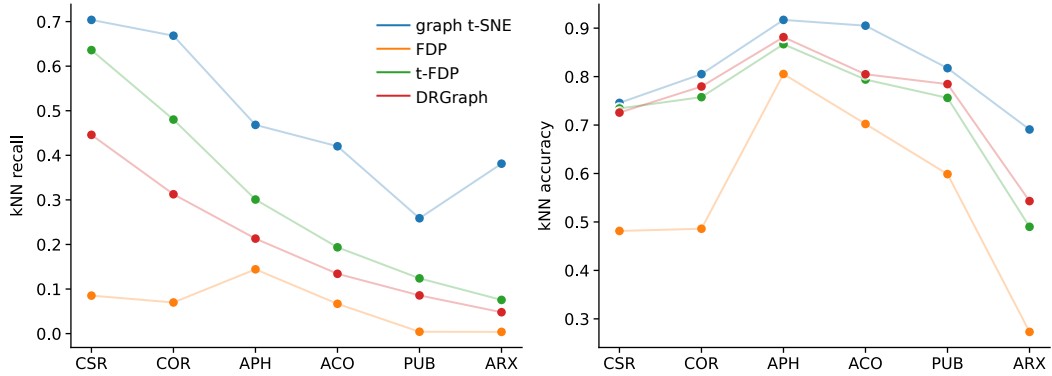

Figure 3: Performance metrics for graph layouts: $k$NN recall and $k$NN accuracy. Datasets are ordered by the increasing sample size. See Figures 2 and A.3 for the corresponding layouts.

are shown in Figure A.1. For the largest dataset (ARX), graph $t$-SNE took around 100 seconds and graph CNE took around 60 minutes.

## 5 GRAPH LAYOUTS VIA GRAPH $t$-SNE

The $t$-SNE algorithm consists of two steps: first, it computes pairwise affinities between all pairs of points based on the $k$NN graph; second, it optimizes the embedding to match these affinities (Section 3.1). For graph $t$-SNE, we replace the first step and obtain the affinity matrix directly from the graph adjacency matrix. We then run $t$-SNE optimization to produce the embedding (Figure 1).

Given an unweighted graph $G = (\mathcal{V}, \mathcal{E})$, its adjacency matrix $\mathbf{A}$ is defined such that $A_{ij} = 1$ if $(i, j) \in \mathcal{E}$ and $A_{ij} = 0$ otherwise. Since all graphs considered in this study are undirected, the adjacency matrix is a binary, symmetric square $n \times n$ matrix. In order to convert it into an affinity matrix suitable for $t$-SNE, we follow the standard $t$-SNE's approach (Section 3.1): divide each row by the sum of its elements, then symmetrize the resulting matrix, and then normalize to sum to 1:

$$\mathbf{P} = \frac{\tilde{\mathbf{A}} + \tilde{\mathbf{A}}^\top}{2n}, \text{ where } \tilde{A}_{ij} = A_{ij} \Big/ \sum_{k=1}^{n} A_{ik}. \tag{7}$$

For optimization, we used the `openTSNE` library (Poličar et al., 2019) with default parameters. It uses Laplacian Eigenmaps (Belkin & Niyogi, 2003) for initialization (Kobak & Linderman, 2021), sets the learning rate equal to $n$ to achieve good convergence (Linderman & Steinerberger, 2019; Belkina et al., 2019), and employs fast FIt-SNE algorithm that has linear $\mathcal{O}(n)$ runtime (Linderman et al., 2019).

We compared graph $t$-SNE with three existing graph layout algorithms: FDP (Fruchterman & Reingold, 1991), DRGraph (Zhu et al., 2020a), and $t$-FDP (Zhong et al., 2023). We chose FDP because it is the default layout algorithm in a popular `NetworkX` package (Hagberg et al., 2008). Two other algorithms, $t$-FDP and DRGraph, are recent and can be considered state-of-the-art (we did not use tsNET (Kruiger et al., 2017) for benchmarking, because it cannot embed large graphs and is outperformed by its successor DRGraph). We used the `NetworkX` implementation of FDP (`networkx.drawing.layout.spring_layout`) and the original implementations of both $t$-FDP and DRGraph, all with default parameters (Figure 2).

We found that graph $t$-SNE consistently outperformed all competitors in terms of both $k$NN recall and $k$NN accuracy (Figure 3): it showed the highest values on all datasets, 12 out of 12 times. In agreement with the original results of Zhu et al. (2020a) and Zhong et al. (2023), we saw that DRGraph and $t$-FDP outperformed FDP in both metrics. Our graph $t$-SNE showed further improvement, and it was substantial: in terms of $k$NN recall, graph $t$-SNE improved on the best competitor on average by 18.2 percentage points, and in terms of $k$NN accuracy — on average by 6.7 percentage points. The improvement was particularly strong for the largest graph (ARX), where performance of other methods strongly deteriorated.

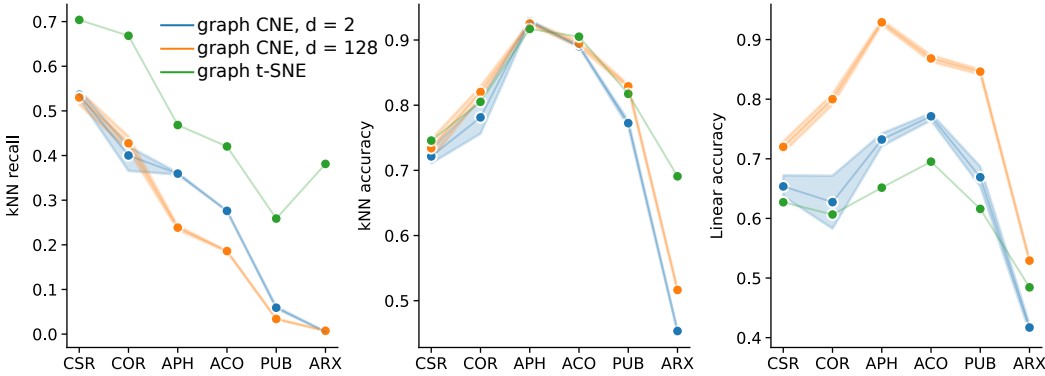

Figure 4: Performance metrics for graph CNE compared to graph $t$-SNE: $k$NN recall, $k$NN classification accuracy, and linear accuracy. Shading shows standard deviation over five CNE runs. Datasets are ordered by the increasing sample size.

Visually, the embeddings produced by graph $t$-SNE looked similar to DRGraph embeddings (Figures 2 and A.3), but showed richer within-class structure, in agreement with the higher $k$NN recall values.

We have also experimented with an alternative way to convert the adjacency matrix into the affinity matrix: namely, to divide $\mathbf{A}$ by the sum of its elements: $\mathbf{P} = \mathbf{A} / \sum_{ij} A_{ij}$. This approach resulted in similar $k$NN recall and $k$NN accuracy values, but gave visually unpleasing embeddings, with low-degree nodes pushed out to the periphery (Figure A.2). Furthermore, we experimented with various initialization schemes, but found that on our graphs, random initialization performed very similar to the default Laplacian Eigenmaps initialization.

## 6 NODE-LEVEL GRAPH CONTRASTIVE LEARNING VIA GRAPH CNE

Similar to $t$-SNE, the CNE algorithm consists of two steps. First, it builds the $k$NN graph of the data. Second, it optimizes the embedding (in our case, parametric embedding) using a contrastive loss function such as InfoNCE to make neighbors be close in the embedding (Section 3.2). For graph CNE, we omit the first step and provide the graph to CNE directly.

We used parametric CNE models, setting the output dimensionality to $d = 2$ and $d = 128$. In both cases we used a fully-connected network (MLP), as is default in CNE, with the number of neurons in each layer $D \rightarrow 100 \rightarrow 100 \rightarrow 100 \rightarrow d$, where $D$ is the number of input node features (Table 1). For both dimensionalities we used the InfoNCE loss. Following Damrich et al. (2022), we used the cosine distance and the Gaussian similarity kernel for $d = 128$, mimicking the standard SimCLR setup (Chen et al., 2020), and the Euclidean distance and the Cauchy similarity kernel for $d = 2$, mimicking the standard $t$-SNE setup. We set the number of negative samples to 100 (increasing it from the default 5 improved the results), and batch size to $\min\{1024, |\mathcal{V}|/10\}$ (in pilot experiments we noticed that small graphs required smaller batch sizes for good convergence). The number of epochs was set to 100. Optimization parameters were left at default values: Adam optimizer (Kingma & Ba, 2015) with learning rate 0.001.

Compared to graph $t$-SNE, graph CNE, with both $d = 2$ and $d = 128$, had lower $k$NN recall (Figure 4). This is likely because graph CNE had to use node features, whereas graph $t$-SNE was unconstrained by them and optimized graph neighborhood preservation directly. At the same time, $k$NN accuracy was very similar (Figure 4) on all datasets, apart from the ARX dataset. The comparatively poor performance of graph CNE on the ARX dataset was likely due to ARX feature space showing weak class separation (Table 2); whereas graph $t$-SNE does not use node features and hence is not influenced by the feature quality. Visually, two-dimensional graph CNE embeddings looked *very* similar to graph $t$-SNE embeddings (Figure 1), even though the former were parametric and the latter were non-parametric.

Table 2: Linear classification accuracy (in %) of graph CNE and existing graph contrastive learning algorithms. Output dimensionality of CNE is indicated in brackets. The line marked by $\star$ shows $k$NN accuracy instead of linear accuracy. CNE values are mean $\pm$ standard deviation across five training runs. Non-CNE values are taken from Zhang et al. (2022), MLP values are taken from `https://openreview.net/forum?id=dSYkYNNZkV&noteId=aLQzIXVy0w` and Guo et al. (2023). OOM denotes out-of-memory error. Datasets are ordered by the increasing sample size. For comparison, the first row shows linear accuracy in the feature space.

| | CSR | COR | APH | ACO | PUB | ARX |
|---|---|---|---|---|---|---|
| *Feature space* | 70.3 | 68.6 | 90.7 | 79.6 | 87.8 | 55.1 |
| Graph CNE (2) | $65.4 \pm 2.2$ | $62.7 \pm 6.2$ | $73.2 \pm 1.4$ | $77.1 \pm 0.7$ | $66.9 \pm 2.3$ | $41.7 \pm 0.8$ |
| Graph CNE (2) $\star$ | $72.1 \pm 1.5$ | $78.1 \pm 3.2$ | $92.9 \pm 0.3$ | $89.0 \pm 0.2$ | $77.2 \pm 0.6$ | $45.3 \pm 0.2$ |
| Graph CNE (128) | $72.0 \pm 1.3$ | $80.0 \pm 1.2$ | $92.9 \pm 0.5$ | $86.8 \pm 0.7$ | $84.6 \pm 0.6$ | $52.9 \pm 0.3$ |
| GRACE | $71.2 \pm 0.5$ | $81.9 \pm 0.4$ | $92.2 \pm 0.2$ | $86.3 \pm 0.3$ | $80.6 \pm 0.4$ | OOM |
| GCA | $72.1 \pm 0.4$ | $82.3 \pm 0.4$ | $92.5 \pm 0.1$ | $87.9 \pm 0.3$ | $80.7 \pm 0.5$ | OOM |
| MVGRL | $73.3 \pm 0.5$ | $83.5 \pm 0.4$ | $91.7 \pm 0.1$ | $87.5 \pm 0.1$ | $80.1 \pm 0.7$ | OOM |
| DGI | $71.8 \pm 0.7$ | $82.3 \pm 0.6$ | $91.6 \pm 0.2$ | $83.9 \pm 0.5$ | $76.8 \pm 0.6$ | $71.2 \pm 0.2$ |
| BGRL | $71.1 \pm 0.8$ | $82.7 \pm 0.6$ | $93.1 \pm 0.3$ | $89.7 \pm 0.4$ | $79.6 \pm 0.5$ | $72.7 \pm 0.2$ |
| CCA-SSG | $73.1 \pm 0.3$ | $84.2 \pm 0.4$ | $93.1 \pm 0.1$ | $88.7 \pm 0.3$ | $81.6 \pm 0.4$ | $72.3 \pm 0.2$ |
| AF-GCL | $72.0 \pm 0.4$ | $83.2 \pm 0.2$ | $92.5 \pm 0.3$ | $89.7 \pm 0.2$ | $79.1 \pm 0.8$ | — |
| AFGRL | $68.7 \pm 0.3$ | $81.3 \pm 0.2$ | $93.2 \pm 0.3$ | $89.9 \pm 0.3$ | $80.6 \pm 0.4$ | OOM |
| Local-GCL | $73.6 \pm 0.4$ | $84.5 \pm 0.4$ | $93.3 \pm 0.4$ | $88.8 \pm 0.4$ | $82.1 \pm 0.5$ | $71.3 \pm 0.3$ |
| Local-GCL, MLP | $70.3 \pm 0.6$ | $78.3 \pm 0.5$ | $90.9 \pm 0.4$ | $82.4 \pm 0.5$ | $79.6 \pm 0.5$ | — |
| GRACE, MLP | $65.5 \pm 2.6$ | $67.7 \pm 0.9$ | $87.9 \pm 0.6$ | $80.9 \pm 1.2$ | $83.3 \pm 0.5$ | — |

As expected, CNE with $d = 128$, yielded considerably higher linear classification accuracy compared to both 2-dimensional embeddings (Figure 4). In terms of linear accuracy, graph CNE performed comparably to the state-of-the-art graph contrastive learning (GCL) algorithms[1] (Table 2). Graph CNE achieved the best results on one of the datasets (PUB), and had close to the best results on other datasets, apart from the ARX.

Note that graph CNE was at disadvantage compared to all other GCL methods listed in Table 2 because it used an MLP network, whereas other GCL methods traditionally use graph convolutional networks (GCN). GCN takes the entire graph as input and uses *message passing*, which pulls together embeddings of connected nodes and helps to obtain better embeddings. However, GCN is not able to transform one node at a time, and so a trained GCN cannot be applied to a new, held-out node. In contrast, our graph CNE with MLP can (after training) process one node at a time, which we consider more appropriate for node-level graph learning (see Discussion). There are very few GCL results based on the MLP architecture reported in the literature. Two examples are Local-GCL and GRACE trained with MLP architecture (reported in the OpenReview discussion of Zhang et al. (2022) and in Guo et al. (2023) respectively, Table 2): both had lower accuracy compared to our graph CNE on *all* datasets.

For the ARX graph, we did not find any existing MLP-based results. Lower performance of graph CNE compared to GCN-based GCL methods was, again, likely due to the feature space of this graph showing only weak class separation (Table 2, first row).

## 7 DISCUSSION

**Summary** Our paper makes three contributions, two practical and one conceptual:

   i. We suggested a novel graph layout algorithm, *graph t-SNE*, and showed that it outperforms existing competitors in preserving local graph structure.

---

[1]We did not measure their performance ourselves, but took the values directly from the literature.

ii. We suggested a novel node-level augmentation-free graph contrastive learning algorithm, *graph CNE*, and showed that it achieves comparable performance to the state-of-the-art methods despite using the MLP architecture, and outperforms existing MLP-based graph contrastive learning results.

iii. We established a *conceptual connection* between graph layouts and graph contrastive learning: we argued that both are instances of graph embeddings (non-parametric 2D embedding and parametric 128D embedding), and both can be efficiently implemented using neighbor embedding frameworks. We suggested a new task, parametric 2D embeddings (Figure 1), as a 'missing link' between these two existing tasks.

**Simplicity**   Both graph $t$-SNE and graph CNE are remarkably simple, because they use existing $t$-SNE and CNE machinery out of the box. This is in stark contrast with competing algorithms. For example, existing graph layout algorithms inspired by $t$-SNE, such as tsNET (Kruiger et al., 2017), DRGraph (Zhu et al., 2020a), and $t$-FDP (Zhong et al., 2023), all develop their own machinery, implementation, and approximations, and deviate from $t$-SNE in many different nontrivial ways (see Section 2). However, as we demonstrated, simply using $t$-SNE (via graph $t$-SNE), outperforms all of them in terms of layout quality.

Similarly, in node-level graph contrastive learning (GCL), the focus has been on developing graph augmentations (see Section 2), following the contrastive learning paradigm in computer vision that is based on image augmentations. Augmentation-free GCL methods such as AFGRL (Lee et al., 2022) and AF-GCL (Li et al., 2023) instead rely on complex heuristics to select positive pairs. Our approach is conceptually much simpler, as it uses the InfoNCE loss function with graph edges as positive pairs, and nothing else. The closest method in the literature is Local-GCL (Zhang et al., 2022), which also uses graph edges as positive pairs. The difference is that Local-GCL uses an approximation scheme to deal with $\mathcal{O}(n^2)$ repulsive forces, whereas we use the standard contrastive learning approach of within-batch repulsion, which is much simpler.

All of the existing GCL methods, including Local-GCL, employ graph convolutional neural networks (GCNs). Recent work argued that the reason many GCL algorithms work well has little to do with the specific augmentations or heuristics they use, but rather is due to their GCN architecture (Trivedi et al., 2022; Guo et al., 2023). GCN uses *message passing* between graph nodes, which implicitly makes representations of connected node pairs more similar. In other words, in GCL algorithms employing GCNs, it is the GCN that does the heavy lifting, and not the specifics of the GCL algorithm. In contrast, our graph CNE uses an MLP network, and nevertheless performed similarly well. See below on why we think MLP is a more suitable choice for node-level GCL tasks.

**Limitations**   In this work, we focused on complex real-world graphs and have purposefully not tested our graph $t$-SNE on simple planar graphs or 3D mesh graphs that are often used for benchmarking graph layout algorithms. We suspect that graph $t$-SNE would perform suboptimally on such graphs, as $t$-SNE is known to have troubles with embedding simple 2D manifolds such as the Swiss roll. To some extent this can be addressed by increasing the degree of freedom parameter of the $t$-distribution or using the Gaussian kernel instead (Kobak et al., 2019), and/or by increasing the exaggeration value (Kobak & Berens, 2019; Böhm et al., 2022; Damrich et al., 2022).

Our graph CNE relies on the MLP and we did not experiment with GCN architecture. This, however, is not a limitation but a purposeful design choice: we think that GCN, whereas very meaningful for graph-level learning, is less applicable for node-level learning, where one may want to apply the trained model to a set of new objects (based on their node features). With GCN, this is not possible, as it requires the entire graph to be passed in at the same time. We therefore consider MLP architecture more appropriate for node-level GCL.

**Take-home message**   We showed that graph layouts and graph contrastive learning are intimately related and can be approached by existing neighbour embedding frameworks, surpassing state-of-the-art results.

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

## A SUPPLEMENTARY FIGURES

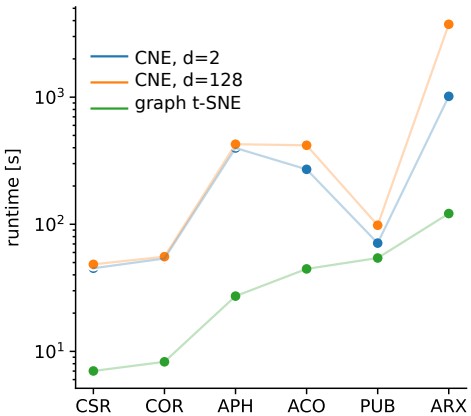

Figure A.1: Computation times for graph $t$-SNE and graph CNE with 2 and 128 output dimensions. openTSNE was run on CPU with `n_jobs=-1`. CNE was run on GPU. Datasets are ordered by the increasing number of nodes. The runtime of openTSNE (for a given number of gradient descent steps) grows linearly with the number of nodes. The runtime of CNE (for a given number of epochs and a given batch size) grows linearly with the number of edges. The Pubmed dataset (PUB) has fewer edges than the Amazon datasets (APH and ACO), see Table 1.

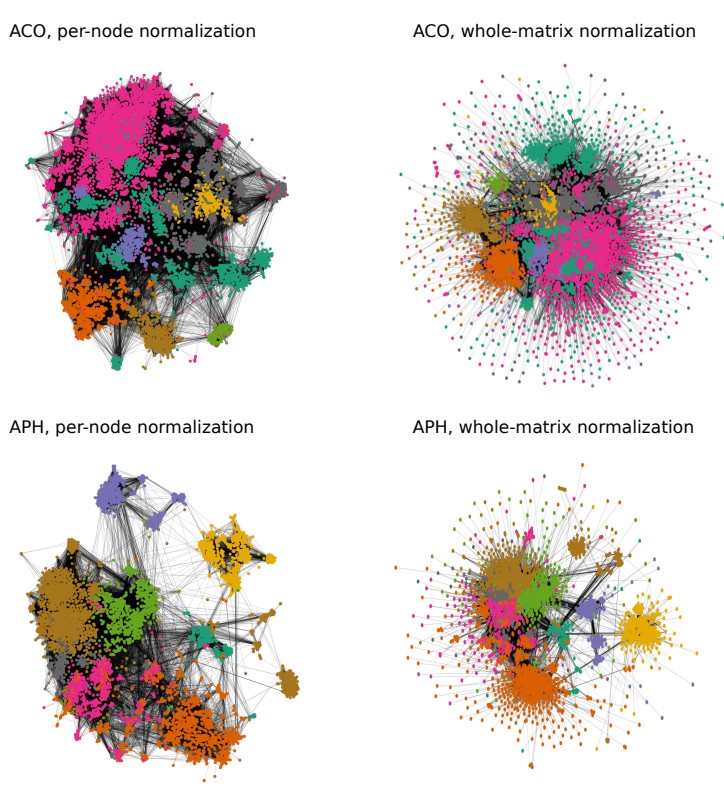

Figure A.2: Graph $t$-SNE visualizations of ACO and APH datasets using per-node normalization of the adjacency matrix (default) and whole-matrix normalization. Embeddings in each row were aligned using Procrustes rotation.

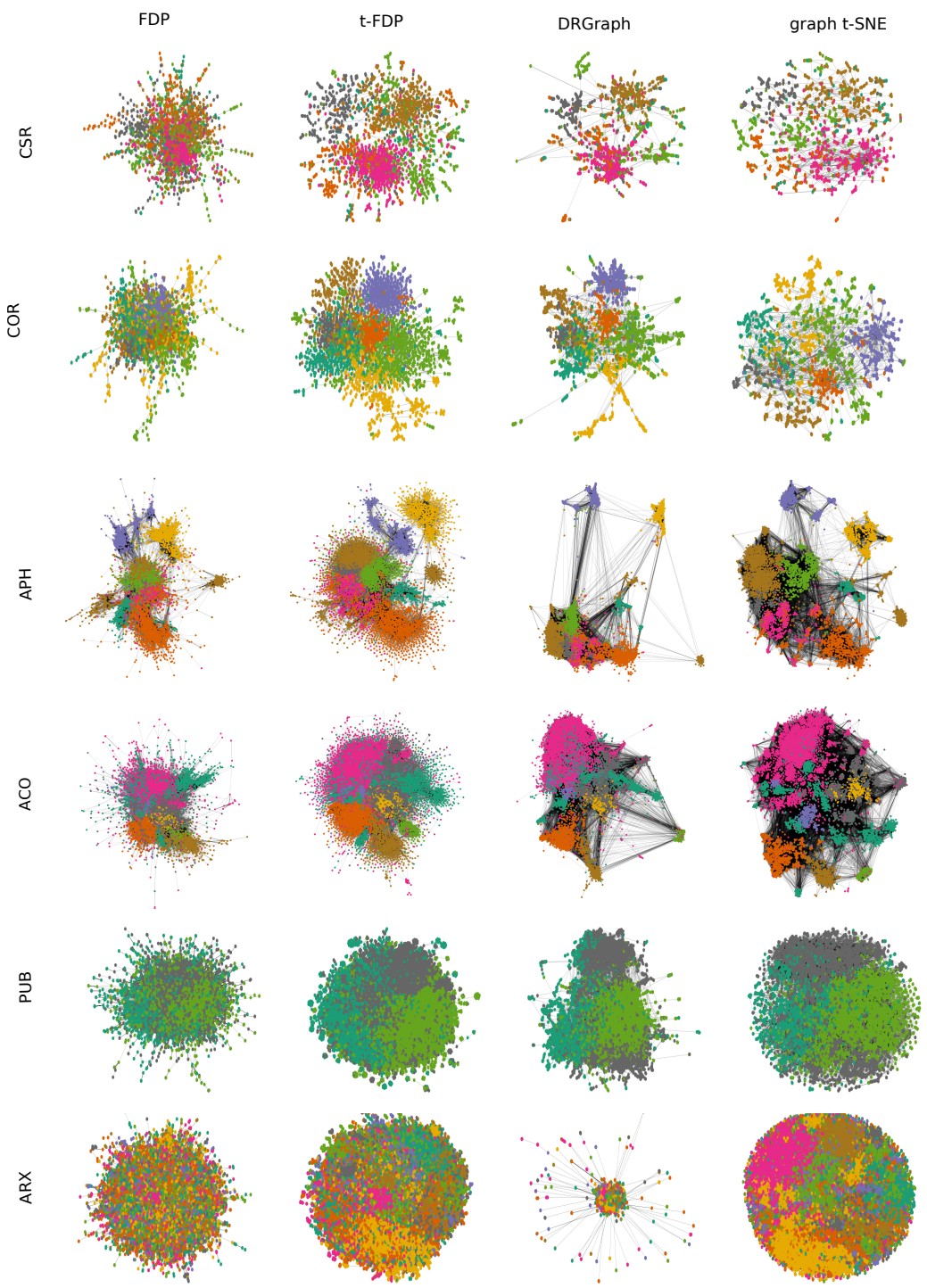

Figure A.3: Embeddings of all considered datasets obtained using FDP (Fruchterman & Reingold, 1991), DRGraph (Zhu et al., 2020a), and $t$-FDP (Zhong et al., 2023), and our graph $t$-SNE. Embeddings in each row were aligned using Procrustes rotation.

