# OpenReview forum: "Graph layouts and graph contrastive learning via neighbour embeddings"
_ICLR.cc/2024/Conference — Submitted to ICLR 2024_

### Official Review · Reviewer_JHfU · 2023-10-31

**Soundness:** 3 good
**Presentation:** 2 fair
**Contribution:** 3 good
**Rating:** 6
**Confidence:** 2

**Summary:**

First, this paper introduces graph t-SNE for two-dimensional graph drawing, and shows that the resulting layouts outperform all existing algorithms in terms of local structure preservation, as measured by kNN classification accuracy. Second, the work introduces graph contrastive neighbor embedding (graph CNE), which uses a fully connected neural network to transform graph node features into an embedding space by optimizing the contrastive InfoNCE objective.

**Strengths:**

1. The experimental performance is good.

2. This method is easy to implement and experimentally efficient.

**Weaknesses:**

1. This work uses MLP to transform node features and gives reasons for not using GCN. If possible, I think it would be helpful to include experiments that use GCN to transform node features.

2. This work seems to apply only at the node level but not at the graph level.

**Questions:**

Is the proposed method applicable at the graph level?

---

> ### Author Response · Authors · 2023-11-16
> **Response**
>
> > The experimental performance is good. This method is easy to implement and experimentally efficient.
>
> We thank the Reviewer for their positive evaluation!
>
> Indeed, our graph t-SNE beats SOTA results by a large margin, and our graph CNE beats SOTA on MLP-based graph contrastive learning. Also, we  believe that the connection between graph layouts and graph contrastive learning is a novel and important insight of our paper.
>
> > This work uses MLP to transform node features and gives reasons for not using GCN. If possible, I think it would be helpful to include experiments that use GCN to transform node features.
>
> We did not use GCNs because all existing GCL papers use GCNs, so we thought it is sufficient to refer to them. For example, Local-GCL is the most similar method to ours but uses GCN. Conveniently, Local-GCL authors also report the results using MLP architecture (quoted in our Table 2), and our Graph CNE outperforms them on **all** datasets.
>
> As we explain in our Discussion, we think that for node-level graph learning, GCNs are not very appropriate, as they cannot process one node at a time. GCNs can be very suitable for graph-level contrastive learning, but this is conceptually  a very different task requiring different datasets (not one graph at a time, but a collection of graphs, such as e.g. molecular stuctures).
>
> We have now found another existing paper that reports MLP-based results (Guo et al., NeurIPS 2023, run experiments on GRACE with MLP: https://arxiv.org/abs/2311.02687), and added it to Table 2. Strikingly, Graph CNE outperforms both reported MLP-based GCL results on **all** datasets.
>
>
> > This work seems to apply only at the node level but not at the graph level. [...] Is the proposed method applicable at the graph level?
>
> Not really. Of course one can average the node-level representation across all nodes and thus obtain a graph-level representation. But we do not think that our method is a reasonable approach for that. To be honest, we do not consider this is a limitation: graph-level contrastive learning is a very different task requiring different datasets (not one graph at a time, but a collection of graphs, such as e.g. molecular structures) and different methods.
>
> In this work we focused on node-level learning, both in 2D (graph layouts) and in high-D (graph contrastive learning).
>
> If there any further concerns, we will be happy to clarify!

---

> > ### Comment · Reviewer_JHfU · 2023-11-22
> >
> > I appreciate the detailed response provided by the authors. I will maintain the original score.

---

### Official Review · Reviewer_WEvC · 2023-11-01

**Soundness:** 2 fair
**Presentation:** 2 fair
**Contribution:** 2 fair
**Rating:** 5
**Confidence:** 4

**Summary:**

The authors aim to achieve both graph layout and node-level graph contrastive learning by neighbor embedding methods. They simplify $t$-SNE and CNE by using graph adjacency matrix as their proposed neighbor embedding methods. Experiments with 3 metrics on 6 public datasets are conducted to demonstrate the effectiveness of the proposed method.

**Strengths:**

1. The results on graph layouts look promising.

2. The proposed graph CNE does not rely on the entire graph as input.

**Weaknesses:**

Please kindly correct me if I misunderstood something.

**Majors:**

1. In Section 4, the authors say "we split all nodes into a training (2/3 of all nodes) and a test set (1/3 of all nodes)." Is this setting applied to all datasets? How to choose nodes for training? Why not use $K$-fold splits?

2. In Section 6, Paragraph 2, the authors say "we used the cosine distance and the Gaussian similarity kernel for $d$=128." Does it make sense to evaluate the models by $k$NN recall/accuracy, which is based on Euclidean distance?

3. In Section 6, Paragraph 3, the authors say "The number of epochs was set to 100." I think the number of epochs should be decided by the convergence of a model. The authors also say "running CNE for more epochs and/or with higher $m$ could likely yield better results" and "be due to insufficient optimization length and/or $m$ value" in the following paragraphs. Does this mean the models are not well-trained?

4. In Section 3.2, the authors use CNE to denote the framework of [1], and the proposed contrastive learning model is named graph CNE. However, the CNE in Figure 4 and Table 2 seems to denote the proposed method, which is confusing.

        [1] Damrich, Sebastian, et al. "From t-SNE to UMAP with contrastive learning." The Eleventh International Conference on Learning Representations. 2022.

5. In Table 2, are the baseline methods using the same experiment settings (for example, train/val/test split) as the proposed method?

6. The authors claim that "graph CNE performed comparably to the state-of-the-art graph contrastive learning algorithms." However, in Table 2, Local-GCL looks better. Please provide more evidence (for example, $t$-test) to support this sentence.

7. Why not try GNN architecture if the authors want to compare the proposed graph CNE with other GCL methods?

**Minors:**

8. Why is Figure 1 shown on Page 1 but cited on Page 7?

9. How are the hyperparameters decided?

10. ".. where running CNE for more epochs and/or with higher $m$ could likely yield better results." Is the $m$ explained somewhere in this paper?

**Questions:**

Please see the weaknesses part.

---

> ### Author Response · Authors · 2023-11-16
> **Response, part 1**
>
> > The results on graph layouts look promising. The proposed graph CNE does not rely on the entire graph as input.
>
> We thank the Reviewer for this assessment!
>
> Just to clarify what we think our main contributions are:
>
> * We are the first to point out and explore conceptual connection between graph layouts and node-level graph contrastive learning, tying together two distinct fields.
> * We suggest 2D graph contrastive learning (graph CNE with $d=2$) as a novel graph learning task as a "missing link" between graph layouts and standard graph contrastive learning.
> * Our graph t-SNE beats SOTA on graph layouts by a large margin (!).
> * Our graph CNE beats SOTA on MLP-based graph contrastive learning.
>
> The Reviewer scored the "novelty" aspect as 1, but we believe the our work contains a whole lot of novel results: both conceptual insights and SOTA achievements!
>
> > In Section 4, the authors say "we split all nodes into a training (2/3 of all nodes) and a test set (1/3 of all nodes)." Is this setting applied to all datasets? How to choose nodes for training? Why not use K-fold splits?
>
> Yes, we used the same 2:1 splitting on all datasets. The set of training nodes was selected randomly. We could indeed use K-fold cross-validation, but used a fixed train/test split for simplicity and because it is standard in the field. The graphs that we used are large enough (number of nodes $n>2000$) so that the sampling error here is small anyway.
>
> We do not see this as a limitation, but if the reviewer disagrees, we are happy to provide some K-fold classification results for comparison.
>
> > In Section 6, Paragraph 2, the authors say "we used the cosine distance and the Gaussian similarity kernel for d=128." Does it make sense to evaluate the models by NN recall/accuracy, which is based on Euclidean distance?
>
> This is an excellent question! Indeed, for $d=128$ embedding we also tried using cosine-metric kNN graph for measuring recall/accuracy, but found that it did not noticeably affect the results. So for simplicity we only reported Euclidean kNN evaluation. We have now inserted a clarification about it into the text.
>
> > In Section 6, Paragraph 3, the authors say "The number of epochs was set to 100." I think the number of epochs should be decided by the convergence of a model. The authors also say "running CNE for more epochs and/or with higher could likely yield better results" and "be due to insufficient optimization length and/or m value" in the following paragraphs. Does this mean the models are not well-trained?
>
> Thank you for bringing this up. Since the original submissions, we have done additional experiments, and by now believe that 100 epochs is sufficient for convergence on all our datasets. We removed all statements that you quoted here.
>
> > In Section 3.2, the authors use CNE to denote the framework of [1], and the proposed contrastive learning model is named graph CNE. However, the CNE in Figure 4 and Table 2 seems to denote the proposed method, which is confusing.
>
> Thank you! We changed Figure 4 and Table 2 to always say "graph CNE".
>
> > In Table 2, are the baseline methods using the same experiment settings (for example, train/val/test split) as the proposed method?
>
> We took the values reported in Table 2 from the literature (as indicated in the caption). The train/test splits there could be different. However, given that the graphs are large and that the train sets are selected at random, that should not play a major role. In fact, the variability due to network re-training (which is what is shown in Table 2 as $\pm$ intervals) is likely larger than the variability due to the test set selection. We have now clarified where exactly the quoted values are coming from.

---

> ### Author Response · Authors · 2023-11-16
> **Response, part 2**
>
> > The authors claim that "graph CNE performed comparably to the state-of-the-art graph contrastive learning algorithms." However, in Table 2, Local-GCL looks better. Please provide more evidence (for example, t-test) to support this sentence.
>
> Indeed, in Table 2 some GCL methods outperform our Graph CNE, but only by a small margin, and on one of the datasets (PUB), Graph CNE is actually the best. This is what we mean by "comparably": not that it performs the best, but it performs close to the best.
>
> Crucially, as we explain in the paper, the more relevant comparison is to the GCL methods when they also use MLP architecture. We have now found another existing paper that reports MLP-based results (Guo et al., NeurIPS 2023, run experiments on GRACE with MLP: https://arxiv.org/abs/2311.02687), and added it to Table 2. Strikingly, Graph CNE outperforms both reported MLP-based GCL results on **all** datasets.
>
> > Why not try GNN architecture if the authors want to compare the proposed graph CNE with other GCL methods?
>
> This is a good question. We did not use GCNs because all existing GCL papers use GCNs, so we thought it is sufficient to refer to them. Local-GCL is the most similar method to ours but uses GCN. Conveniently, Local-GCL authors also report the results using MLP architecture (quoted in our Table 2), and our Graph CNE outperforms them on all datasets.
>
> As we explain in our Discussion, we think that for node-level graph learning, GCNs are not very appropriate, as they cannot process one node at a time. For graph-level learning, GCNs can be very appropriate, but that is another story.
>
> > Why is Figure 1 shown on Page 1 but cited on Page 7?
>
> We added a reference to Figure 1 on Page 2, and also in some other places throughout the text. Thank you.
>
> > How are the hyperparameters decided?
>
> For graph t-SNE, we used all default hyperparameters of openTSNE.
>
> For graph CNE, we mostly used default hyperparameters of CNE, but increased the number of epochs and the number of negative samples to ensure convergence and improve the InfoNCE approximation (as per Damrich et al. 2023). We also adapted the batch size to ensure convergence, based on our pilot experiments.
>
> We added some clarifications to the text.
>
> > ".. where running CNE for more epochs and/or with higher m could likely yield better results." Is the m explained somewhere in this paper?
>
> Yes, the $m$ value is explained in Section 3.2 (see Equation 4), but we now removed this sentence because we now believe that $m=100$ that we used was sufficiently high.
>
> If there any further concerns, we will be happy to clarify!

---

> > ### Comment · Reviewer_WEvC · 2023-11-21
> > **Response to the Rebuttle**
> >
> > I appreciate the authors' efforts in providing a detailed response and an updated submission, which addresses some of my concerns, so I upgraded my rating to weak reject. However, I still have major concerns in the contrastive learning experiment settings (Q1, Q5, and Q7) as follows:
> >
> > It is not fair to compare the proposed method with the baselines if they are not done under the same experiment settings.
> >
> > **For training/val/test splits**, the proposed method "split all nodes into a training (2/3 of all nodes) and a test set (1/3 of all nodes)," but the baseline methods are not. The Citeseer/Cora/Pubmed datasets have public splits, and the APH/ACO datasets are split as 1:1:8 by some of the baselines, which is significantly different from the proposed method.
> >
> > **For architecture**, the proposed method adopts MLP and many other baselines use GCN. It is good if the proposed method can outperform others, however, it does not. I suggest the authors explore the GNN architecture to showcase the superior performance of the proposed method under similar settings compared to others, and then elucidate the trade-off issues associated with employing MLP.

---

> > > ### Author Response · Authors · 2023-11-23
> > >
> > > Thank you for the response and for raising your score!
> > >
> > > > The Citeseer/Cora/Pubmed datasets have public splits, and the APH/ACO datasets are split as 1:1:8 by some of the baselines, which is significantly different from the proposed method.
> > >
> > > Thank you very much for emphasizing this point again. We did not realize until now that the public splits (as well as 1:1:8 splits) only use a small number of nodes for training, whereas we used 2/3 of the nodes. This does indeed make the comparison shown in Table 2 unfair.
> > >
> > > Unfortunately we did not have time anymore to change our evaluation within the response period (ending in a few hours). We will definitely change the evaluation in future revisions, and in case the paper does get accepted here, we will make sure to use the standard evaluation for the final version.
> > >
> > > **Please note that this does not affect our results on graph layouts. It also does not affect our more conceptual results on the relationship between graph layouts and graph contrastive learning, as illustrated by 2D graph CNE shown in Figure 1.**
> > >
> > > > For architecture, the proposed method adopts MLP and many other baselines use GCN. It is good if the proposed method can outperform others, however, it does not. I suggest the authors explore the GNN architecture to showcase the superior performance of the proposed method under similar settings compared to others, and then elucidate the trade-off issues associated with employing MLP.
> > >
> > > Thank you for re-iterating this concern. As we said, here we feel that https://arxiv.org/abs/2311.02687 (NeurIPS 2023) has already done exactly that, and while it may be interesting to repeat similar experiments to corroborate their results, this can be arguably seen as not really novel anymore.
> > >
> > > We do not claim that our method with GCN architecture would outperform the existing methods, this is not the point of our paper. We did not check that, and do not know if it is the case. But it may indeed be interesting to run these experiments, and we will look into it.

---

### Official Review · Reviewer_pX5U · 2023-11-05

**Soundness:** 2 fair
**Presentation:** 2 fair
**Contribution:** 2 fair
**Rating:** 5
**Confidence:** 3

**Summary:**

The paper proposes a new algorithm for graph layouts, graph t-SNE, and a new algorithm for contrastive learning algorithm, graph
CNE, and draw a connection between the two algorithms.

**Strengths:**

1. The paper uses several real-world graph datasets to demonstrate the superiority of the proposed methods.
2. The exploration of the connection between graph layouts and graph contrastive learning is a relatively under-explored yet important topic given the abundance of graph data nowadays and the need to visualize and process graph data.

**Weaknesses:**

1. Paper writing can be made better. For example, for Equation 3, there is no description of the meaning of y. The definitions of parametric and non-parametric embeddings should be clearly described earlier in the paper.
2. As the authors mention, it is suspected that "in GCL algorithms employing GCNs, it is the GCN that does the heavy lifting, and not the specifics of the GCL algorithm." However, there is no experimental setup where the authors verify this hypothesis, which is a pity.

**Questions:**

1. Does graph t-SNE use the graph node features? This is asked because for Graph CNE, the paper mentions the reduction of dimensionality from D (input feature dimension) to d (2 or 128), and to my understanding, all the methods for graph layouts use only the structure/topology of the graph. If this is the case, I am still unsure if kNN classification accuracy is the right metric, since the metric quantifies local class separation, yet the class label for each node is unobserved by the layout methods. If this is the case, please justify the adoption of such a metric for comparing different graph layout algorithms. The concern is, what if for a real-world dataset, the class labels of nodes are less related or unrelated to the structure/topology of the graph? Then wouldn't all the layout methods show low accuracy? Fundamentally, the question is about the justification of using this metric to evaluate layout algorithms.

---

> ### Author Response · Authors · 2023-11-16
> **Response**
>
> > The exploration of the connection between graph layouts and graph contrastive learning is a relatively under-explored yet important topic given the abundance of graph data nowadays and the need to visualize and process graph data.
>
> We thank the reviewer for this assessment!
>
> Indeed, we also believe that this connection is conceptually very important, and our paper is the first to point it out and explore. Along the way, we beat SOTA on graph layouts, beat SOTA on MLP-based graph contrastive learning, and suggest 2D graph contrastive learning (graph CNE with $d=2$) as a novel graph learning task and a "missing link" between graph layouts and standard graph contrastive learning.
>
> > Paper writing can be made better. For example, for Equation 3, there is no description of the meaning of y. The definitions of parametric and non-parametric embeddings should be clearly described earlier in the paper.
>
> Thank you for these suggestions! We added the definition of $\mathbf y_i$ vectors into Section 3.1 and added a definition of parameteric/non-parametric embeddings into Section 3.2. We also carefully went over the entire paper and inserted clarifications and cross-references. Please do let us know if you find any other specific places that should be clarified!
>
> > As the authors mention, it is suspected that "in GCL algorithms employing GCNs, it is the GCN that does the heavy lifting, and not the specifics of the GCL algorithm." However, there is no experimental setup where the authors verify this hypothesis, which is a pity.
>
> Indeed, we do not provide experimental evidence for this, but in fact, another paper (not ours) has just been accepted to NeurIPS and published online: Guo et al. 2023 Architecture matters: Uncovering implicit mechanisms in graph contrastive learning (https://arxiv.org/abs/2311.02687). This paper explicitly studies the effect that GCN architecture has in GCL and indeed shows that it does the heavy lifting in pulling connected nodes together. We have therefore reformulated this paragraph and cited this paper.
>
> > Does graph t-SNE use the graph node features?
>
> No it does not. The Reviewer's understanding is correct.
>
> > If this is the case, I am still unsure if kNN classification accuracy is the right metric, since the metric quantifies local class separation, yet the class label for each node is unobserved by the layout methods. If this is the case, please justify the adoption of such a metric for comparing different graph layout algorithms. The concern is, what if for a real-world dataset, the class labels of nodes are less related or unrelated to the structure/topology of the graph? Then wouldn't all the layout methods show low accuracy? Fundamentally, the question is about the justification of using this metric to evaluate layout algorithms.
>
> This is an excellent question. Indeed, the standard approach in the graph layout literature is to use kNN preservation metrics, such as the kNN recall that we use in Figure 3a. We also consider it our main metric for graph layouts (and note that our graph t-SNE outperformed all other methods on all considered datasets, according to this metric).
>
> We additionally showed kNN accuracy in Figure 3b because it is similar to the metric always used in node-level graph contrastive learning (linear classification accuracy). Given that one of the aims of the paper is to establish the conceptual connection between graph layouts and graph contrastive learning, we think it is useful to show this metric here. But again, we agree with the Reviewer that the kNN recall metric is more important here.
>
> We have clarified this in the text.
>
> If there any further concerns, we will be happy to clarify!

---

> ### Author Response · Authors · 2023-11-23
>
> Dear reviewer pX5U, thanks again for your feedback. We hope our edits have resolved your concerns! If there are any remaining concerns, we would still have 2 hours to respond.

---

### Meta-Review · Area_Chair_4BHk · 2023-12-06

**Metareview:**

The paper introduces two new algorithms: graph t-SNE for graph layout and graph CNE for contrastive learning. Graph t-SNE is designed for two-dimensional graph drawing and is shown to outperform existing algorithms in preserving local graph structure, as measured by kNN classification accuracy. Graph CNE uses a fully connected neural network to transform graph node features into an embedding space by optimizing the contrastive InfoNCE objective.

Strengths:
* It explores the connection between graph layouts and graph contrastive learning, which is a relatively under-explored area, and this can be valuable given the increasing importance of graph data.
* The proposed graph CNE does not rely on the entire graph as input, which can be advantageous in some scenarios.

Weaknesses:
* There is a lack of experimental setup to verify the hypothesis that the success of graph contrastive learning algorithms primarily depends on GCNs and not the specifics of the GCL algorithm.
* Questions are raised about the choice of kNN classification accuracy as a metric for evaluating graph layout algorithms, especially when class labels may not be related to graph structure.
* Some sections of the paper appear to have inconsistencies or unclear naming conventions, which may confuse readers.
* The paper does not explore the use of GCN for transforming node features, which could be an interesting comparison.

**Justification For Why Not Higher Score:**

The weaknesses of the paper include writing improvements needed for clarity, the lack of experimental verification of a key hypothesis, questions about the choice of evaluation metric, naming inconsistencies, and the absence of more comprehensive experiments comparing the proposed method with GCN-based transformations for node features.

**Justification For Why Not Lower Score:**

N/A.

---

### Decision · Program_Chairs · 2024-01-16

Reject